



# Annual variation characteristics of Eurasian hydrologic elements and their linkage with climate and environment changes during 1951-2015

Jia Qin [1], Yongjian Ding [2,3], Tianding Han [2], Junhao Li [1], Shaoping Wang [2], Yaping Chang [1]

[1] Key Laboratory of Eco-hydrology Inland River Basin, Northwest Institute of Eco-Environment and Resources, Chinese Academy of Sciences, Lanzhou, 730000, China
[2] State Key Laboratory of Cryospheric Sciences, Northwest Institute of Eco-Environment and Resources, Chinese Academy of Sciences, Lanzhou, 730000, China
[3] University of Chinese Academy of Sciences, Beijing, 100049, China

*Correspondence to*: Jia Qin (qinjia418@lzb.ac.cn)

**Abstract.** In this paper, the variations of the lowest monthly discharge (LD), mean monthly discharge (MD), and highest monthly discharge value (HD) during 1951-2015, as well as spring snowmelt water and winter river ice change, in eleven major rivers, distributed respectively in the high-latitudes (55°N -70°N), middle latitudes (40°N -55°N), and lower latitudes (30°N -40°N) of Eurasia, were analysed. Energy and water budgets in different watersheds were compared to detect the

reasons for Eurasian hydrological changes. We found that the annual LD in most Eurasian rivers was increasing since the 1950s, with rates of (5%-8%) per decade. But the increase rate slowed down after the late 1990s in the middle latitudes of Eurasia. Both the MD and HD in the lower latitudes of Eurasia had increasing trends during 1951-2015, while they had little changes in the high and middle latitudes. The river ice thickness and volume have been continuously reducing since the 1950s, as well as the maximum snow water equivalent. And ice period of the Eurasian rivers has shortened about 24 days.

The LD trend is mostly dominated by temperature via impacting river ice thickness and extent, while the HD is mostly impacted by snowmelt water and rainfall respectively in different latitudes. Annual MD trend is controlled by evapotranspiration, especially after the late 1990s. After the late 1990s, a 'warm Arctic-large discharge' pattern existed in the lower and high latitudes of Eurasia, but a 'warm Arctic- few discharge' pattern in the middle latitudes (except the winter).
Keyword: Eurasian river discharge; river ice; snowmelt water; surface energy balance; climate change; Arctic warming

## 1 Introduction

Freshwater plays an important role in the hydrological cycle of the ocean because it is essential for the maintenance of low-salinity surface water layer and formation of sea ice (Matthiessen et al., 2000). Eurasian rivers, such as the Ob River, the Lena River, the Yenisei River, etc., contribute about approximately 80% of total freshwater recharged into the circum-Arctic epicontinental seas every year (Gordeev et al., 1996). These rivers also can stabilize and effectively supply production, living

and ecological water demands, and have an important impact on nutrients migration in Eurasia (Jia et al., 2017). A large amount of the organic matter in the Arctic Ocean comes from the older degraded material in the major Arctic rivers (e.g.

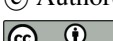



Kolyma, Lena, Ob, and Yenisei) (Stedmon et al., 2011). With global warming, land surface involving hydrological processes were significantly impacted (Labat et al., 2004; Berg et al., 2016).

Some hydrological variables in Eurasian basins, such as evapotranspiration, runoff, melt-water, and permafrost Freeze-thaw process, changed significantly in the past decades (Ding et al., 2006; Wang et al., 2009; Jaeger et al., 2011). In recent years, there are a lot of discussions about Eurasian water balance with respect to its response to climate change (McClelland et al., 2004; Liu et al., 2014; Zhou et al., 2015; He et al., 2018). The existing researches effectively enhanced the development of Eurasian land-surface hydrology, and can help to systematically analysis the Eurasian surface hydrological change during the past decades, especially after the late 1990s when the global and Eurasian climate significantly changed

(Magnuson et al., 2000; McClelland et al., 2004), although most of them focused on specific watershed or regions in Eurasia (Smith et al., 2007; Tananaev et al., 2016; Duan et al., 2017).

The physical mechanism driving the variations of Eurasian river discharge and other hydrologic factors continue to fuel debate. Changes in temperature, precipitation, permafrost, reservoirs, fire frequency, and plant transpiration have all been posited, and some results are different, or even diametrically opposite (Zhang et al., 2000; Serreze et al., 2002; Peterson et al.,

2002; McClelland et al., 2004; Gedney et al., 2006; Walvoord and Striegl, 2007; Ye et al., 2009; Shiklomanov et al., 2013; Grosse et al., 2016). There has large difference of water-energy balance in different latitudes. So river discharge variations and the dominate influence factors in different latitudes of Eurasia may be quite different. In addition, the impact scope and extent of Arctic climate on Eurasia and its changes add the complexity of hydrology variations (Easterling et al., 2000; Chen et al., 2011). This study selected eleven major rivers in Eurasia to analyze the variations of hydrologic factors (including

river discharge, river ice and snowmelt) and the zonal differences during 1951-2015. Spatial and temporal variations of water and energy budgets, as well as Arctic index were used to systematically analyze the reasons for Eurasian hydrological changes in the past 65 years, so as to further understand the mechanism of Eurasian hydrologic variations.

## 2 Materials and methods

In the study, Annual discharge in eleven large rivers (hydrological stations) of Eurasia was analyzed. The detailed

information of these rivers was shown in Table1. According to the latitude of each river station, they were divided into three groups: high-latitudes (55°N -70°N), middle latitudes (40°N -55°N), and lower latitudes (30°N -40°N). Long time series of river discharge in these rivers were collected from the Global Runoff Data Centre, 56068 Koblenz, Germany (2007) (http://www.bafg.de/GRDC/EN/Home/homepage_node.html) and the national hydrological stations of China. The hydrologic data in most of these rivers ranges from 1951 to 2015, with the shortest time series (57 years) in the Shulehe

River (Changmabu station) of China. These rivers are located in different regions of Eurasia, and their watershed area controlled by the hydrological stations shown in Table 1 range from 10961 to 2950000 square kilometers. In addition to river discharge, the changes of Eurasian river ice also were analyzed in the text. The trends of river-ice thickness during 1934-2012, as well as the freeze and breakup dates in different Eurasian rivers were compared. These relative river-ice data was





downloaded from the Arctic Data Center (https://arcticdata.io/), and they were used in analyzing river-ice inter-annual

variation and the river-ice phenology in Eurasia. The volume of river ice during river water freezing period of each

hydrologic year was calculated by the following Eq. (1) and Eq. (2) (Brooks et al., 2013):

$$V_{ice} = \sum_{i=1}^{n} h_i \times A_i \times 10^{-12}, \tag{1}$$

$$h_i = \alpha \times \left(D_{f(i)}\right)^{1/2} \tag{2}$$

where $V_{ice}$ is the annual peak ice volume of each river (km3), and $A_i$ is the channel area of each river (m2), $h_i$ is peak river-

ice thickness of different grid (mm), $D_{f(i)}$ is the sum of accumulated freezing degree days (℃) in river typical ice period (1

November to 30 April) of a specific hydrologic year, and α is an ice growth coefficient, which were $20mm \cdot ℃^{-1/2}d^{-1/2}$ ,

according to the existing study result (Brooks et al., 2013), n is the grid numbers of channels in a river. The gridded daily air

temperature data employed to calculate $D_{f(i)}$ were obtained from CERA-20C, a gridded climatology data set (spatial

resolution of 0.5°) covering the period November 1950 to April 2010 (Laloyaux et al., 2016).

Snowmelt water is an essential component of water resource, and plays an important role in regional and global water

cycles (Jiancheng et al., 2016). About 80% area of Eurasia and North America is covered by snow in winter (Robinson et al.,

1993). In the study, the maximum snow water equivalent (SWE) in late winter and early spring were used to quantitative

evaluate snowmelt runoff in Eurasia. The snowmelt water in a specific month is a simple calculation of the maximum SWE

in the specific month minus that last month. The SWE data were downloaded from Terrestrial Water Budget Data Archive:

Monthly Time Series (1900 - 2017) (Version 4.01) (Willmott et al., 1985).

Surface energy budgets in different latitudes of Eurasia were analyzed to detect the reason for runoff changing in

Eurasia. Each surface energy budget in the lower, middle and high latitudes of Eurasia was calculated from the daily ERA

reanalysis datasets (http://apps.ecmwf.int/datasets/data/interim-full-daily/). In addition, some climate data, soil temperature

and moisture data, as well as Arctic Oscillation index were used to analyze the causes for the Eurasian hydrologic variation.

The lowest monthly discharge (LD), mean discharge (MD) and the highest monthly discharge (HD) in each calendar year

during 1951-2015 in different Eurasian rivers were respectively analyzed in the text. We comprehensively considered the

snowmelt water, precipitation, temperature, evapotranspiration, the hydrothermal variation in permafrost active layer, as well

as extreme climate data to analyze the HD and MD changes in the summer half of a year. According to the observed river

discharge data, the monthly LD in Eurasian rivers mostly occurs in winter, and the HD occurs in the summer half of a year.

In an attempt to detect the reason for Eurasian LD changes during 1950-2015, the correlations between LD and winter air

temperature (mean, maximum, and minimum), which can influence land surface snowmelt, surface soil ice and river ice in

warm days of winter, as well as previous autumn rainfall and soil hydrothermal (September-October), which impact the

quantities of wintertime groundwater flow out, were comparatively analyzed using SPSS software. The linear regression

coefficients of dominated variables were calculated by the stepwise method by simultaneously removing those not important.



The time series of precipitation, air temperature, and soil temperature data in each river basin were obtained from the interpolation results using the ordinary Kriging method, based on the Land-Based Station Data of the National Climatic Data Center (NCDC) (https://www.ncdc.noaa.gov/). Gridded soil moisture data were obtained from Terrestrial Water Budget Data Archive: Monthly Time Series (1900 - 2017) (V. 4.01) (Willmott et al., 1985), and were calculated and interpolated by the Matlab 2016a to watershed scale. Gridded extreme climate data (e.g. the maximum and minimum air temperature) were

obtained from the Climdex indices (https://www.climdex.org/) (Donat et al., 2013).

## 3 The results

### 3.1 Variations of annual discharge in Eurasian rivers

Figure 1 shows annual variations of the LD, MD and HD in the eleven rivers during 1951-2015. The change rates of LD in Eurasian rivers are significantly larger than MD and HD during the past 65 years, no matter in the high latitudes or low

altitudes. The LD was increasing from 1951 to 2015, with the rate of (5%-8%) per decades in the Eurasian rivers. And the increase rate of LD in rivers located in low latitudes was larger than that in the middle and higher latitudes. In addition, it was obviously that LD in middle-Eurasia had larger fluctuation than that in rivers of other two groups, according to the determination coefficients (R2) in Fig. 1. The increasing rate of LD in the middle altitudes was decreased from 9.6% (1951-1995) to 5.1% (1951-2015) per decade. It slowed down and turned to a slight decreasing trend after 1996 (Fig.1). The MD

and HD had similar variation trend during the past decades (Fig. 1). They all increased in rivers of the high latitudes and low altitudes, but had a slight decline in the middle-latitude river basins during 1951-2015.

### 3.2 River ice changes in the past decades

River-ice phenology is sensitive to regional climate change, and the river-ice quantity can effectively predict spring flood size in specific river basins (Magnuson et al., 2000). Figure 2 shows the freeze and breakup dates of river ice in seven major

Eurasian rivers (50°N -70°N). The consistent trends in different rivers provide evidence of later freezing and earlier breakup of river ice in Eurasia from 1951 to 2012 (Fig. 2). Over the 62 year, changes in breakup dates ranged 0.10-0.37 days per year earlier, and changes in freeze dates ranged 0.02-0.28 days per year later. The earlier trend of breakup dates and the later trend of freeze dates were significantly speeding up after the 1990s (Fig.2). Ice breakup dates averaged 14.8 days earlier in spring, and the freeze dates averaged 9.5 days later in autumn from 1951 to 2012. The freezing period of these rivers has been

significantly shortened about 24 days since the 1950s. The long-term trends in observed river ice phenology provide evidence that winter season below freezing temperature was shortening over the past decades. The phenomenal of earlier breakup dates in spring and later freeze dates in autumn was consistent with the result analyzed using the data of 1846-1995 (Magnuson et al., 2000), while the speed of ice-period shortening during 1996-2012 was intensified, compared to that in 1846-1995. In addition, river ice thickness was reducing from 1951 to 2012 in the Eurasian rivers (Fig. 3), with decreasing

rate 2.8-1.1 cm per decade. River-ice volume was also continuously reduced from 1951, with the total ice volume for the




major Eurasian rivers decreasing by 5.48 km$^3$ (8.2%) over the 1951-2010 records (Fig. 4). These long-term and changed trends in observed river ice phenology provide evidence that Eurasian river discharge are responding to climate warming trends, and they increase confidence in the patterns of Eurasian climate warming over the past decades, especially over the past 20 years.

**3.3 Snow water equivalent changes**

According to Fig. 5, the annual maximum SWE in the middle and high latitudes of Eurasia was both decreasing from 1951 to 2014. In the high latitudes, there are three obvious turning points (late-1960s, early-1980s, and late-1990s) dividing the SWE time series (1951-2014) into four time periods. And it is clear that the increasing or decreasing trend of spring SWE is very similar to that of the summer HD in each time period in the high latitudes of Eurasia though comparing the Fig. 5 and

Fig. 1. By contrast, the fluctuation of the SWE in the middle latitudes of Eurasia was gentle during 1951-2014. According to the quantities of the calculated snowmelt water, the major snow-melt period of a year is April-May in the middle latitudes of Eurasia, while it in the high latitudes of Eurasia is May-June (Fig. 6). Snowmelt water contributed about 50±8.8mm water to the land surface during April-May in the middle part of the Eurasian, and about 112±18.7mm water to the high latitudes of Eurasia in May-June.

**3.4 The reason for river discharge variations considering energy and water balance**

**3.4.1 The reason for winter LD change**

As mentioned above, LD in Eurasian rivers was continuous increasing during the 1951-2015. To clear the reason for the LD variation, energy and water balance were analysed in the study. Each surface energy budget in the lower, middle and high latitudes of Eurasia had the similar variation trend in winter (Fig.7). The surface latent heat flux (LH) in Eurasia was

increasing with a rate of 0.22 W m$^{-2}$ decade$^{-1}$ from 1980 to 2015, while the sensitive heat flux (SH) was decreasing with a rate of 0.51 W m$^{-2}$ decade$^{-1}$. The decreased surface net solar radiation (Rs) (0.23 W m$^{-2}$ decade$^{-1}$) and slightly increased surface thermal radiation (Rt) (0.06 W m$^{-2}$ decade$^{-1}$) made surface net radiation (Rn) decrease (0.27 W m$^{-2}$ decade$^{-1}$). The soil heat flux (Hs) in Eurasian river basins, especially in the HL and ML of Eurasia, had little changes (0.01 W m$^{-2}$ decade$^{-1}$) during 1980-2015, calculated by the land surface Energy Balance Equation (the energy flux of Photosynthesis was ignored).

The little changed Hs means that the heat which permafrost obtained from or released into the atmosphere was stable in most Eurasian river basins and the thawing-freezing thermal condition of permafrost almost had no changes during 1980-2015, which made the effect of permafrost on hydrology (such as the water flow path in permafrost active layer) changed little, and the impact of water from permafrost ice melt on the river discharge variation also can be ignored in that period.

The effects of climate and soil hydrothermal conditions in winter, as well as the former seasons (late summer and

autumn), on the LD were further analyzed. As shown in Table 2, the R$^2$ (coefficients of determination) in the major Eurasian river basins is ranged from 0.281 to 0.634, based on multiple linear regression analysis of annual LD (dependent variable)



and different possible impact factors (independent variable) during 1951-2015. It means that the dominated factors can explain 28.1%-63.4% of the variability in annual LD during the past decades. The winter air temperature has significantly positive correlations with LD in most Eurasian rivers (Table 2). The air temperature was a major controlling factor for the

LD variation in Eurasia. By contrast, the effect of previous hydrothermal conditions on the winter LD variations in most Eurasian rivers during 1951-2015 is not too significant.

Lower winter air temperature makes river ice freeze deeper, which could reduce river flow in a river through increasing channel water storage and diminishing, even cutting off groundwater inflow in some channel system (Prowse and Beltaos, 2002). In contrast, high winter temperature could result in larger winter river discharge. The "contribution" of decreased

river ice to winter LD increase in each Eurasian river (except for the Yangtze River), calculated by the slope of annual LD divided by the slope of river-ice volume during ice period (1 November to 30 April) from 1951 to 2010, ranged from 6.7% to 41.5%, with the mean value of 19.6%. It means that river-ice decrease is an important reason for Eurasian LD variation during the past decades, and it should not be ignored. After the 1950s, winter air temperature in most part of Eurasia was rising (Cohen et al., 2014), leading to river-ice thinner and ice volume significantly decrease (Fig. 3 and Fig. 4), thus the

winter LD continuously increasing (Fig. 1). Note that the rising trend of LD over the mid-latitudes of Eurasia has stopped since the 1990s (Fig. 1), when the winter air temperature there exhibited a cooling trend (Cohen et al., 2014).

### 3.4.2 The reason for HD and MD change

The HD of each river in the middle and high latitudes of Eurasia (except the Selenga River) always occurs in May or June, when large amount of snowmelt and rainfall flow into river channels. The spring SWE and HD in the middle latitudes of

Eurasia present a same decreasing trend during 1951-2015 (Fig. 1 and 5). We calculated that the "contribution" of the decreased snowmelt water to HD decrease in each river of the middle latitudes of Eurasia ranged from 57.3% (Kyzyl station in the Yeniesei River) and 61.5% (the Biya River Basin) to almost 100% (the Selenga and Volga River Basin), based on the slope of SWE and HD changes in each river during 1951-2015. In contrast, the HD in the high latitudes of Eurasia changed little during 1951-2015, but the SWE was decline significantly in the same period (Fig. 1 and 5). Spring snowmelt floods

were occurring earlier in the Eurasian river basins in the past decades (Gautier et al., 2018). The speed up of snow melting could forms more frequent and larger spring floods, and this in some extent would offset the reducing effect of SWE on HD in the high latitudes of Eurasia. The role of spring snowmelt on HD variation in the high latitudes of Eurasia also should not be ignored. In addition, precipitation of May-June in the middle and high latitudes of Eurasia both had little changes during the past decades (Fig.8). The correlation coefficient of precipitation (May-June) and HD in the middle and high latitudes of

Eurasia are smaller than 0.1 (P>0.1). It is concluded that the changes of snowmelt water perhaps determined the HD variations in the middle and high latitudes of Eurasia during 1951-2015. The HD in the lower latitudes always occurs in July-September, and it is always effectively recharged by the heavy rainfall and glacier melt water in this period (Ding et al., 2006; Jia et al., 2017). The speed up of total precipitation and extreme precipitation increase (Fig.9) can partly explain why the HD trend in the lower altitudes of Eurasia suddenly changed after the late 1990s.



190       The MD trend in each latitudes of Eurasia was similar to the HD in the past decades (Fig.1). Over 80% of annual discharge in Eurasian rivers is formed in the warm season (May-October) (Gautier et al., 2018), and the annual MD process is mainly determined by summer river discharge. The total precipitation in the lower latitudes of Eurasia had an increase trend during 1951-2015, especially after the late-1990s (Fig.9), which is similar to the MD variation in the same region. The correlation coefficient ($R^2$) of precipitation and MD is 0.31 (significant level <0.01). It means 31% of annual MD variation

could be explained by total precipitation in the lower altitudes of Eurasia. Figure 10 shows the energy budget variation in the summer half of each year. From 1980 to late-1990s, the evapotranspiration of the lower latitudes of Eurasia had an increase trend as the rising LH (Fig.10), while precipitation in this period had little change (Fig.9). In addition, the Rs decrease in this period made permafrost cooling, thus reduced melt water of permafrost contained ice release. Water consumption by increased evapotranspiration and the declined river runoff recharged by permafrost ice melt water caused MD decline during

this period (Fig.1). on the contrary, the MD rising trend after the late 1990s was mainly caused by the increasing precipitation and lower evapotranspiration in the lower latitudes of Eurasia, according to the precipitation, LH and Rs variations (Fig 9 and 10). According to Fig. 9, annual precipitation in the high latitudes of Eurasia had continuously increase trend during 1951-2015, while winter snowfall there was decreasing (Fig.5). It means the summer rainfall was increasing during 1951-2015. The MD in this region had a slight increase trend, but its slope during 1951-2015 was obvious smaller

than that of MD (Fig.1). This means other factors reduced the runoff recharge effect of precipitation in the high latitudes of Eurasia. We compared the surface energy balance during 1980-2015 (Fig. 10), and found that the LH was significant increasing with a rate of 0.72 W m$^{-2}$ decade$^{-1}$, while the SH, Rn and Hs were all decreasing, with rate of 1.18, 0.13, 0.59 W m$^{-2}$ decade$^{-1}$ respectively. Larger LH means more land surface evapotranspiration, and thus reduce the river discharge in some extent. The decreasing Rn, which mainly caused by surface longwave radiation increase, lead to land surface energy

outcome larger and the Hs reduce, thus reducing the permafrost ice melt via decrease the thawing thermal of permafrost. So it can be concluded that the slightly increase of MD in the high latitudes of Eurasia was the comprehensive effect of the increased precipitation, larger evapotranspiration consumption, and the reduction of permafrost ice melt. The contribution of increased precipitation to water resource in this region was almost consumed by the increased evaporation. To the middle latitudes of Eurasia, the MD during the 1951-2015 was decreasing, but it had an obvious increasing trend after the late 1990s.

Precipitation there almost had no change before the late-1990, while it turned to quickly increase after the late-1990 (Fig.9). The LH in the middle latitudes had the fastest rising rate during 1979-2000, compared to that in other regions of Eurasia, and Hs has slight decrease in this period. After the late 1990s, LH suddenly turned to a continuous decreasing trend. More plentiful precipitation (Fig.9) and less evaporation led to MD in the middle latitudes of Eurasia increase after the late 1990s.



## 4 Discussion

### 4.1 The possible impact of Arctic warming on Eurasian hydrological variations

The Arctic was warming significantly in recent decades, with a land-surface air temperature rise rate of 0.3-0.4°C per decade during 1961-2004 (Chapin et al., 2005), and even a linear trend of 0.755 °C per decade during 1998–2012 (Huang et al, 2017). The Arctic has warmed approximately twice times more rapid than the entire northern hemisphere during the past few decades (Screen and Simmonds, 2010; Serreze et al., 2009), a phenomenon called Arctic Amplification. This amplification would cause water vapor transfer of Eurasia continent to be changed and lead to an increased probability of extreme weather events (Francis and Vavrus, 2012). Observational evidence shows that significant cold anomalies over the Eurasia in winter are associated with the decrease of the Arctic sea-ice cover in the preceding summer-to-autumn seasons and the negative phase of the North Atlantic Oscillation (Honda et al, 2009). Less sea ice and increased open water in Arctic also lead to higher Arctic atmosphere moisture, which is advected southward to impact Eurasian precipitation (Cohen et al., 2014). In the past decades, the frequency of climate extremes in Eurasia occurred much higher (Min et al., 2011; Cohen et al., 2014).

Existing researches shows that three broad dynamical frameworks: storm tracks mainly in the North Atlantic, jet stream, and Planetary waves, can explain how the Arctic change, including the changes of Arctic sea ice and relative boundary layer in the Arctic, impacting on mid-latitudes (e.g. Eurasia) (Cohen et al., 2014). Overland et al., (2011) found that a 'warm Arctic-cold continents' pattern existed in winter. This pattern is suitable for Eurasia in the period after the late 1990s. During this period, the Arctic warming significant speeded up in winter with the middle latitudes of Eurasia (40°N -55°N) obvious cooling, thus LD decreasing via river ice increasing in thickness and extent. While in the lower latitudes (30°N -40°N) and high latitudes (55°N -70°N) of Eurasia, the LD increasing speed were both faster, because of less river ice caused by rising surface air temperature, which was consistent with Arctic Warming. Gong and Ho (2003) found that the Arctic Oscillation (AO) has close negative relationships with East Asian summer monsoon based on the correlation analysis in 1900-1998. A positive phase of the AO leads to a northward shift in the summertime upper tropospheric jet stream over East Asia, which is closely related to anomalous sinking motion in 20°-40°N and rising motion in surrounding regions. These changes give rise to a drier condition over the region extending from the Yangtze River valley to the southern Japan and a wetter condition in the southern China (Gong and Ho, 2003). Figure 11 shows the summer AO variations during 1950-2015. We found that the 'AO positive phase-dry Asia' was also very applicable after the late-1990s, when most summer AO index were negative, while summer precipitation and discharge in this period was significant rising in 30°N -40°N (Fig.1 and 9). According to simulations of an atmospheric model, the Arctic sea ice loss induces a southward shift of the summer jet stream over Europe and increased European precipitation (Screen, 2013). It can be concluded that Eurasia hydrological factors, especially the river discharge in winter has good response to the Arctic climate and sea ice change. While in summertime, the impact of Arctic on river discharge is more obvious in the lower latitudes of Eurasia.



### 4.2 The impact of natural factor vs. human activity on Eurasian hydrology


Currently, there exists a lot of debates in impact mechanism of hydrological variations in Eurasia, including natural and artificial factors (McClelland et al., 2004; Francis and Vavrus, 2012; Cohen et al., 2014). It is ascertained that these hydrological factors was impacted by Eurasian climate involve weather extreme changes, which has close relationships with Arctic climate and sea ice change, even though the effects were varied or opposite in different time periods. With the global

and Arctic warming since the late 1990s, relative surface hydrology significantly changed, and the energy-water balance have obvious seasonal and zonal differences in Eurasia. So to simply predict Eurasian hydrological change based on global surface air temperature variation (Peterson et al., 2002) is sometimes questionable. Establishing a reasonable linkage between Arctic change and Eurasian climate and weather to furtherly forecast the Eurasian hydrological response may be more credible.

In addition, dams and permafrost also have impacts on discharge of Eurasian rivers, especially Eurasian arctic rivers (McClelland et al., 2004). Numerous large dams have been built in Eurasia (Malik et al., 2000). Dams have dramatically altered the seasonality of discharge but are not responsible for increases in annual values, although there also a viewpoint of controlling the seasonality of flow could have a positive or negative influence on long-term trends in annual discharge (Charles et al., 1997). Permafrost may have contributed to changes in discharge of Eurasian rivers, but it cannot be

considered a major driver (McClelland et al., 2004). This ascertained that natural factors were the controlled reason for Eurasian hydrologic variations in the past decades.

### 5 Conclusions

The trend of LD, MD and HD was varied in different latitudes of Eurasia during 1951-2015. Based on the river stations and their climate characters, the paper divided three latitude zones: high-latitudes (55°N -70°N), middle latitudes (40°N -55°N),

and lower latitudes (30°N -40°N), to analyse the hydrological variations and the reasons. LD in almost all Eurasian rivers had an increase trend, but the increase rate in the lower latitudes has slow down since the late 1990s. Under the background of Arctic warming, Eurasia precipitation and temperature had significantly response. This Eurasian climate change made river ice and snowfall corresponding changed, thus influence the river discharge. During 1951-2015, the ice breakup occurs earlier about 14.8 days in spring, and the freeze period in autumn averaged 9.5 days later from 1951 to 2012. The ice period

was significantly shortened (24 days) and river ice thickness and ice volume were also reducing since 1950s in the Eurasian rivers. Winter river ice reduction in thickness and extent with temperature warming played an important role in LD increasing during 1951-2015. While after the late 1990s, winter river ice thickness in the middle latitudes of Eurasia was increasing with the air temperature (especially the Tmin) significantly cool under a 'warm Arctic-cold continents' pattern, thus the winter LD decrease.

The MD and HD in the lower latitudes of Eurasia both had increasing trend caused by increased precipitation, especially the plentiful summer rainfall during 1951-2015. While in the high and middle latitudes, the MD had slight



changes. Although Arctic warming and sea ice reduction had led to precipitation increasing there, the evapotranspiration also increase significantly (because of surface latent heat increasing), combining with permafrost water-heat effect. Snowmelt water contributed about 50±8.8mm water during April-May of each year to the land surface in the middle part of the

Eurasian, and about 112±18.7mm water to the high latitudes of Eurasia in May-June. SWE stored in winter was decreasing from 1951 to 2014 in the middle and high latitudes of Eurasia. The decreased snowmelt was a determined factor for HD decline in the middle latitudes of Eurasia during the past decades, as May-June precipitation in the middle latitudes of Eurasia almost had no change. Based on the comparisons of Arctic climate and Eurasia hydrological changes, it can be concluded that a 'warm Arctic-large discharge' pattern existed in the lower and high latitudes of Eurasia, but a 'warm Arctic-

few discharge' pattern in the middle latitudes (except the winter) after the late 1990s.

**Author contribution**

Jia Qin and Yongjian Ding developed the idea and outlines of the article. Jia Qin prepared the manuscript with contributions from all co-authors.

**Competing interests**

The authors declare that they have no conflict of interest.

**Acknowledgements**

This work was supported by the National Natural Science Foundation of China (Grant No. 41877156, 41730751, and 41771040).

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





**Table 1: Eleven selected river basins in Eurasia and the hydrological station information.**

|  | **Hydrological station** | **River** | **Latitude** | **Longitude** | **Catchment area (km$^2$)** | **Runoff series** |
|---|---|---|---|---|---|---|
| High altitudes | Malonisogorskaya | Mezen | 65.03 | 45.62 | 56400 | 1951-2014 |
|  | Salekhard | Ob | 66.57 | 66.53 | 2950000 | 1951-2015 |
|  | Kuz' movka | Yenisei | 62.32 | 92.12 | 218000 | 1951-2015 |
|  | Krestovskoye | Lena | 59.73 | 113.17 | 440000 | 1951-2015 |
|  | Verhoyansk | Yana | 67.58 | 133.42 | 45300 | 1952-2015 |
| Middle altitudes | Volgograd power plant | Volga | 48.80 | 44.59 | 1360000 | 1953-2010 |
|  | Biysk | Biya | 52.55 | 85.28 | 36900 | 1951-2015 |
|  | Kyzyl | Yenisei | 51.72 | 94.40 | 115000 | 1951-2015 |
|  | Mostovoy | Selenga | 52.03 | 107.48 | 440200 | 1951-2014 |
| Lower altitudes | Changmabu | Shule | 39.82 | 96.85 | 10961 | 1953-2009 |
|  | Hankou | Yangtze | 30.58 | 114.28 | 1488036 | 1951-2008 |

* Note: The catchment area is the hydrometric basin area upstream of the station and not the total basin area.





**Table 2: The correlations between winter LD and different variables in typical river basins of Eurasia**

| Variables ⟍ Baseflow | ST | | Moisture$_{au}$ | Rainfall$_{au}$ | T$_a$ | | | R$^2$ (Sig.<0.05) |
|---|---|---|---|---|---|---|---|---|
| | ST$_{au}$ | ST$_w$ | | | Mean | Max | Min | |
| Mezen | | | | | | | * | 0.313 |
| Ob | | | | | ** | | * | 0.293 |
| Yenisei | | | | ** | * | | | 0.558 |
| Lena | | | * | | | | ** | 0.281 |
| Yana | | *** | | | ** | | * | 0.365 |
| Biya | * | | ** | | | | | 0.342 |
| Selenga | | | | | | ** | * | 0.411 |
| Shulehe | | | ** | *** | * | | | 0.634 |
| Yangtze | | | | | | ** | * | 0.470 |

ST$_{au}$ represents soil temperature in previous autumn, ST$_w$ represents soil temperature in winter, Moisture$_{au}$ and Rainfall$_{au}$ represents soil moisture and rainfall in previous autumn respectively, T$_a$ represents air temperature in winter; *represents that is the first control factor of LD variation; ** and *** represent the second and third impact factors of winter LD variation; R$^2$ is the coefficients of determination.






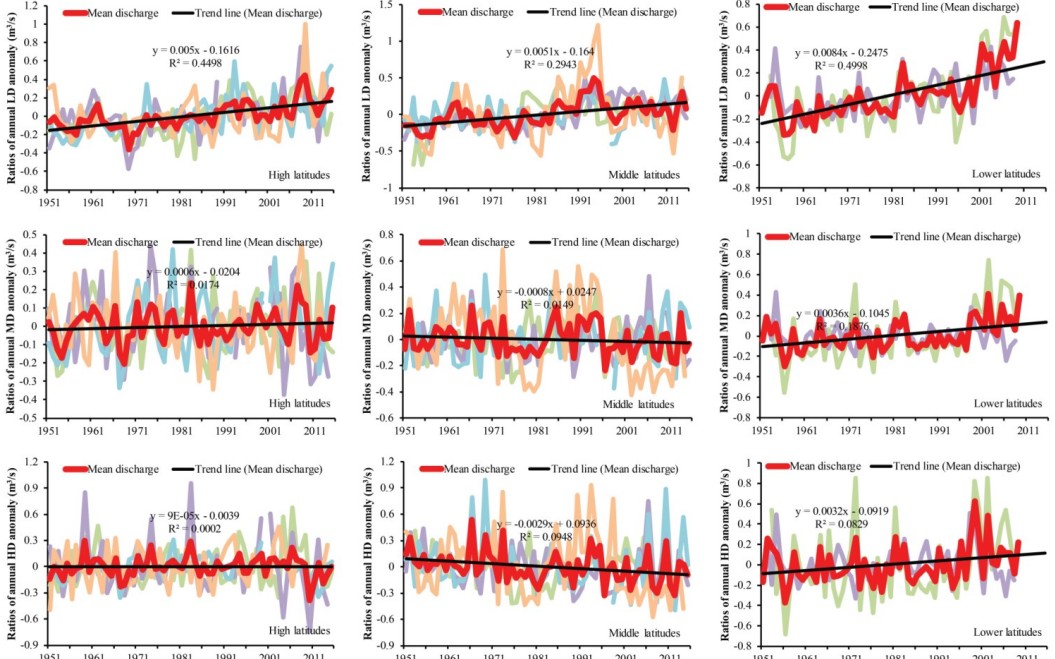

**Figure 1: Time series (1951-2015) of the lowest monthly discharge (LD), mean discharge (MD) and the highest monthly discharge (HD) in major Eurasian rivers (shown in Table 1) lied in the high latitude (55°N -70°N), middle latitude (40°N -55°N), and lower latitude (30°N -40°N)), respectively. The thicker line with red color in each small chart represents the mean value of the annual discharge in the different rivers (shown with other color).**






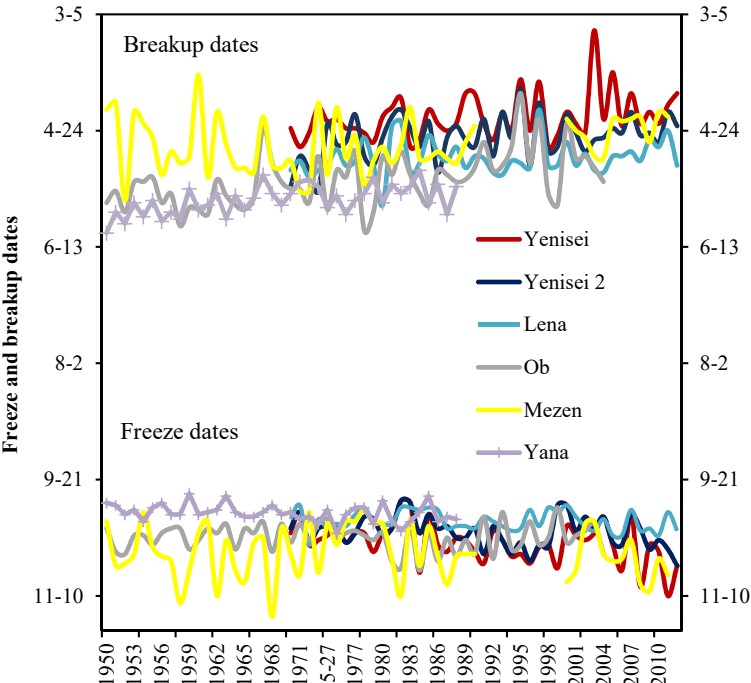

**Figure 2: Time series (1950- 2012) of freeze and breakup dates in Eurasian rivers (55°N -70°N).**

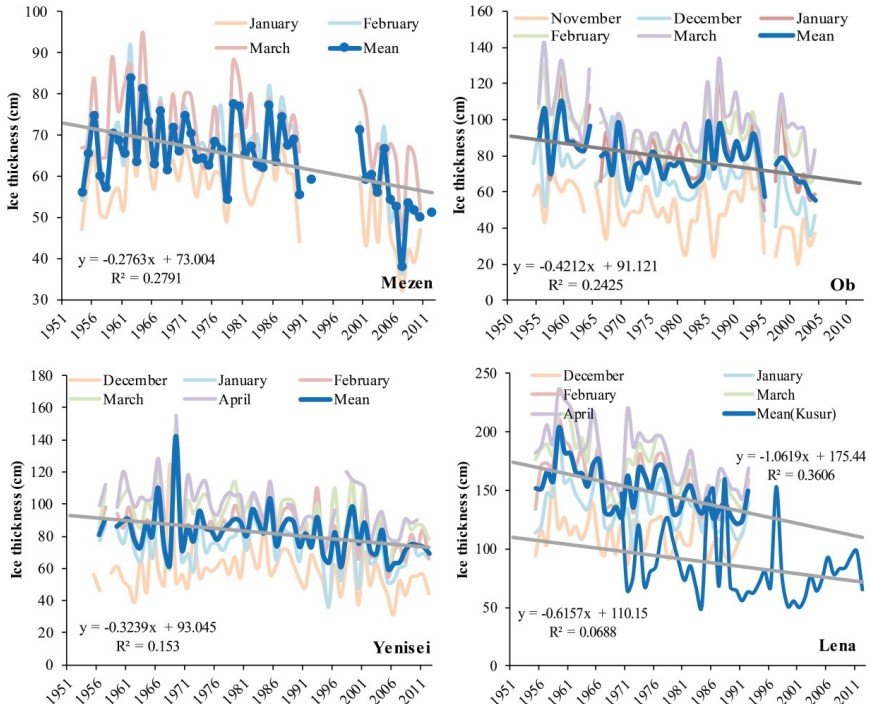

**Figure 3: Variations of winter ice thickness in different Eurasian rivers during 1951-2012.**





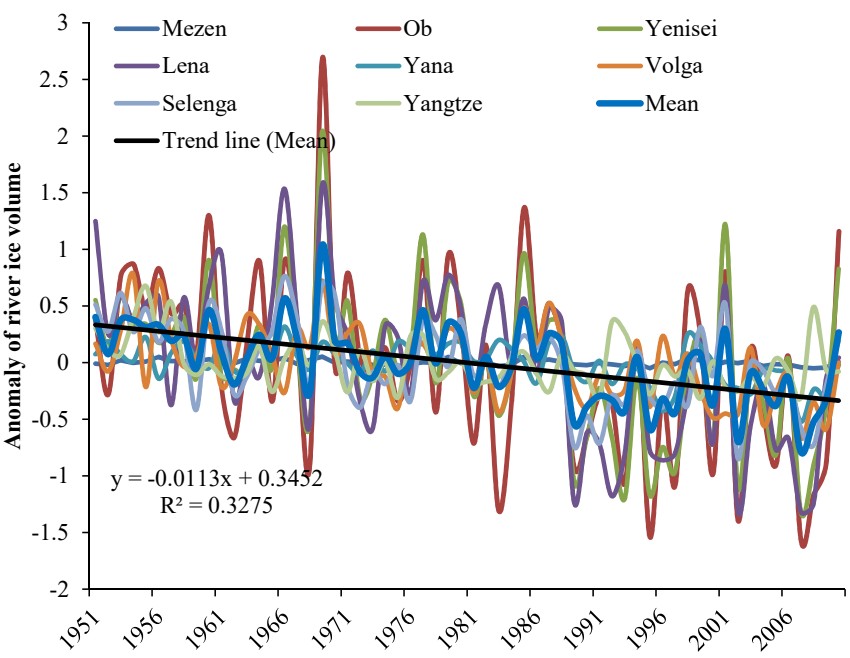


**Figure 4: Anomaly of annual peak river-ice volume (km3) during 1951-2010 in the typical Eurasian rivers.**

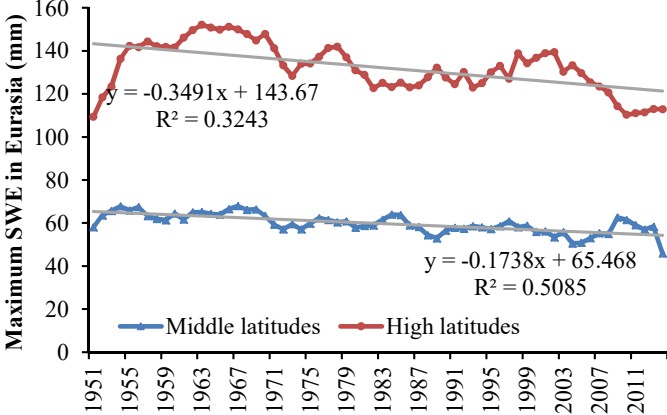



**Figure 5: The maximum snow water equivalent (SWE) (5-year running averages) in late winter and early spring during 1951-2014 in Eurasia (the significant level P<0.01).**

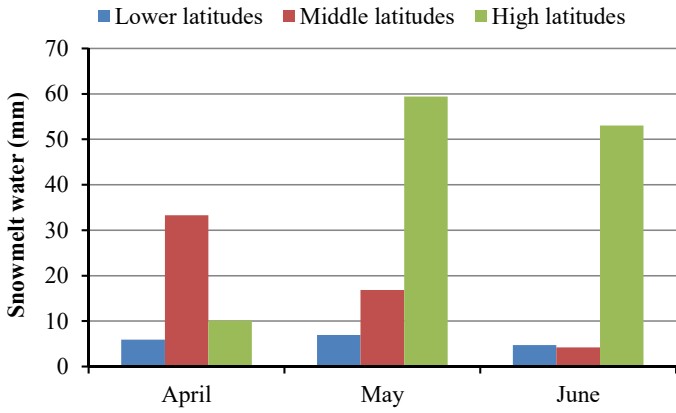


**Figure 6: Mean value of monthly snowmelt water in spring during 1951-2014 in different latitude zones of Eurasia calculated by the difference of monthly SWE.**

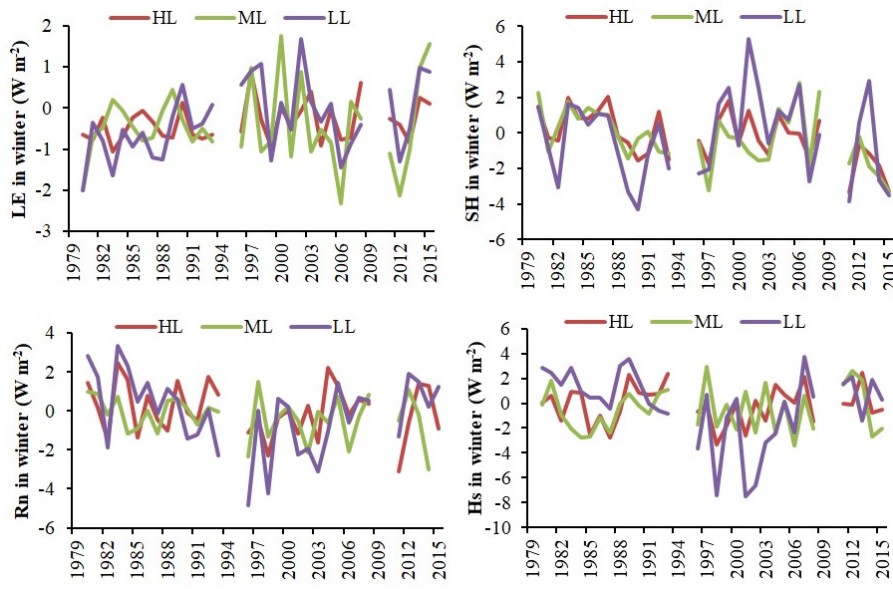

**Figure 7: Winter surface energy budgets in the high latitudes (HL), middle latitudes (ML), and the lower latitudes (LL) of Eurasia**
**during 1980-2015.**



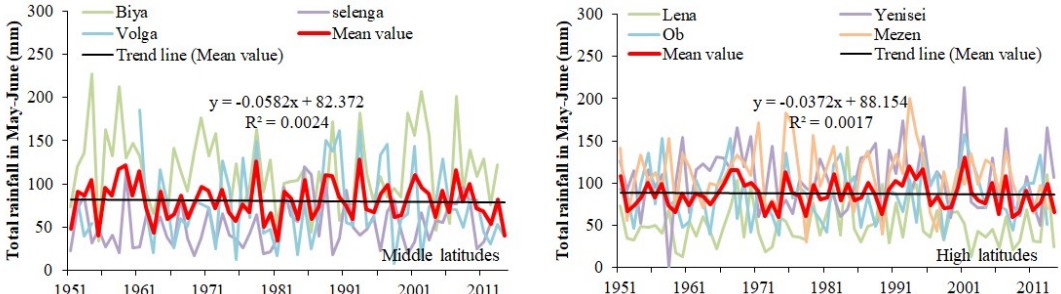

**Figure 8: Total rainfall in May-June of 1951-2014 in the middle and high latitudes of Eurasia.**

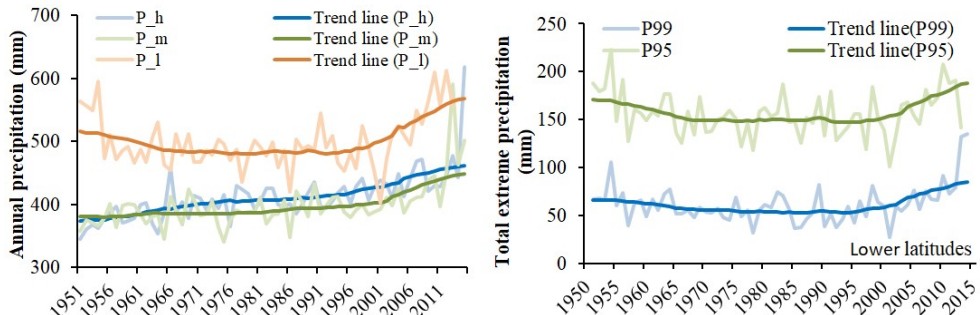

**Figure 9: The precipitation in different latitudes of Eurasia. The P_h, P_m, and P_l is the annual precipitation in the high latitudes, middle latitudes, and lower latitudes of Eurasia, respectively. P95 and P99 are the very wet-day precipitation, that represent precipitation during days exceeding the 95th and 99th percentiles. The trend line is Gauss curve of each variable.**





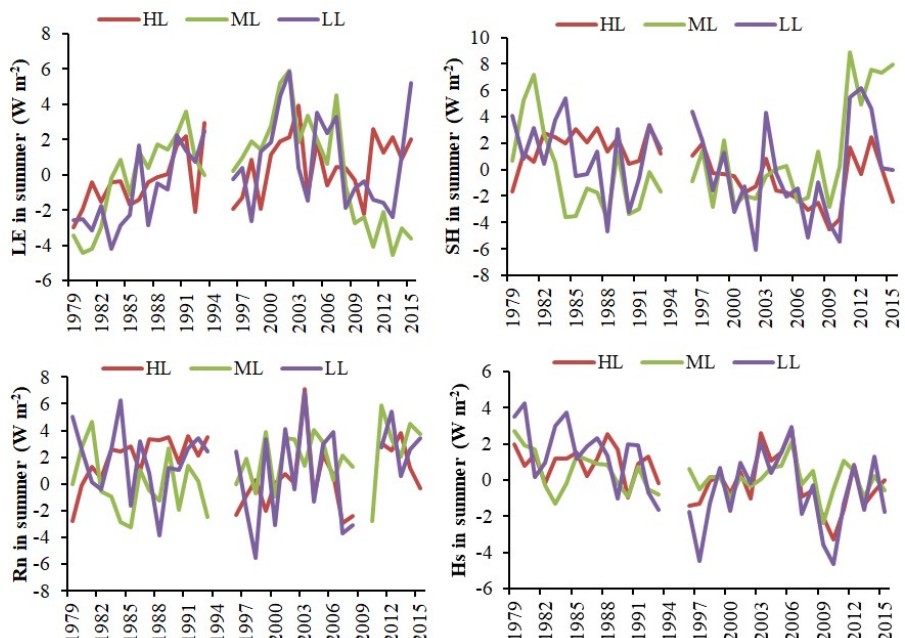

**Figure 10: Surface energy budgets of the summer half year in the high latitudes (HL), middle latitudes (ML), and the lower**
**latitudes (LL) of Eurasia during 1980-2015.**

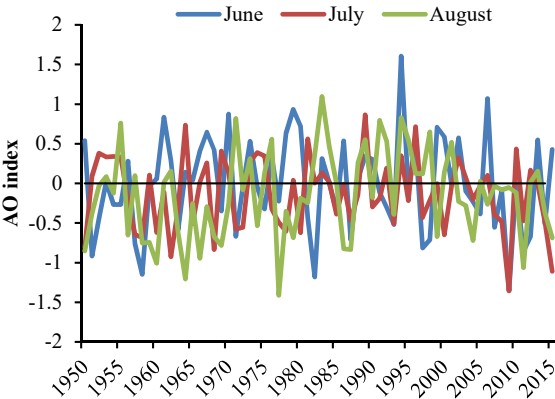

**Figure 11: Time-series of summer Arctic Oscillation (AO) index.**