# Peer review of "Annual variation characteristics of Eurasian hydrologic elements and their linkage with climate and environment changes during 1951-2015"

_Hydrology and Earth System Sciences, 2019_

## Referee Comment (RC1) · Anonymous Referee #1 · 8 Jan 2020

The authors have performed substantial analyses on a number of datasets to try to say something about variations over the last several decades in hydroclimate, as represented by the low, mean, and high discharges measured in a number of Eurasian river basins. Overall I applaud their efforts, but I found the exposition a bit difficult to follow and not always convincing. I've reviewed hundreds of papers over the years, and this was, for me, one of the most difficult I've had to review – not based on content, but based on the presentation. There is good science in here, but it's hidden. I recommend major revision before this paper is accepted.

1. The English is poor, enough to reduce significantly the effectiveness of the discus-

sion. Substantial editing would be needed – not just in terms of correcting individual words but also in terms of the phrasing of arguments. I had a very difficult time getting through some of the longer paragraphs in the results and discussion sections.

2. Much of the data used here is not described well, and appropriate caveats or qualifications are missing. By not discussing the weaknesses of the datasets, the authors are essentially implying that they are all accurate enough for the analyses performed. This may or may not be true. a. What is the rain gauge density underlying the precipitation product? The gauge availability in high latitudes would probably be insufficient for particularly accurate data. Caveats are needed. b. The SWE data are said to come from the Terrestrial Water Budget Data Archive. Some description is necessary. Is this a model product? SWE is notoriously difficult to measure from space, and in situ measurements are presumably not comprehensive. Model products (including reanalyses) are going to be error prone as well, so it's hard to believe that available data will be highly accurate. Again, qualification is needed. c. Data for surface energy budget terms and for soil moisture and temperature are taken from daily reanalysis. I see from the website quoted that these are Interim data rather than ERA5 data; this should be spelled out. Also, appropriate caveats regarding the accuracy of these data, particularly in areas that aren't well measured, are needed.

3. The concept of "trend" is used too loosely in this paper. Typically a trend refers to somewhat consistent changes over multiple decades. Despite what's stated on lines 109-111, I don't see any obvious trend for MD and HD in high latitudes (certainly not a statistically significant trend), and in low latitudes, a trend is seen for MD only if the final ten years are included – before that, there's no trend at all (i.e., for all we know, we could be looking at decadal-scale variability). (Also, it's very strange that the middle latitude MD and HD time series is described as a "slight decline" when this trend seems so much stronger than that for the high latitude MD and HD time series.) Line 123 says that the speed of ice-period shortening during 1996-2012 was intensified compared to that happening earlier, but from Figure 2, except for the Yana, it's not obvious that

there was a statistically significant trend before 1996. By eye, it's not clear that there's any significant trend in Figure 7, despite the statement on line 144 – plenty of decadal variability, though. Again, talking in terms of trend rather than decadal variability is perhaps reading too much into the data. In any case, significance testing for trends is needed throughout. (I did see some p values listed here and there.)

4. The discussion on p. 7 (lines 190-218) was especially difficult to wade through, and I wasn't especially convinced by the arguments – for me, they came off as convoluted. I read through this text several times and am still not clear on the arguments. The discussion regarding permafrost, for example, comes off as unnecessary speculation. Perhaps breaking the paragraph down into more digestible bites would be useful. Here the concept of trend seems especially unclear, with discussion, for example, of an increasing LH trend up to 2000 and a decreasing trend thereafter (lines 216-217). It makes sense to talk about this in terms of decadal variability, not trends. (Also, in Figure 10, why are there breaks in the data in 1994 and 2009? Aren't the reanalysis data complete?)

Minor points

– A map is needed to show where the river basins are located and which ones are included in the high/middle/low latitude categories. This will help the reader understand how representative these basins are for Eurasia in general.

– People will be confused by the term "Arctic-few discharge" in the abstract and on line 290.

– Line 73: Is the CERA-20C dataset really a climatology dataset, or do the values vary from year to year? I assume the latter. Are these reanalysis data?

– Line 78: It probably should be stated explicitly here that sublimation impacts on the snow water balance are neglected. This could be a questionable assumption, as estimates in the literature seem to suggest that sublimation accounts for ∼10-20% of

the snow water balance.

– Line 93: What is SPSS software?

– Line 105 and throughout the text: "altitudes" to "latitudes"?

– I don't understand the slope ratio calculation on lines 165-166. Why not just look at $R^2$ to determine how river ice to winter LD covary? This would make a lot more sense. The same comment applies to the calculation described on line 178.

– Line 197: Rs to Hs?

– Lines 204-205: I don't understand this sentence; there must be a typo.

―――――――――――――――

---

## Referee Comment (RC2) · Anonymous Referee #2 · 1 May 2020

This manuscript analyzed the variations and the zonal differences of hydrological elements in Eurasia, and their causes during the recent few decades. I found the analysis results are interesting and believe that it would add useful information on water resource changes in middle and high latitudes of Northern Hemisphere. However, in my opinion, some revisions to the manuscript are needed, most of which are editorial, but also including the addition of some explanatory text.

—— Section "Introduction" It well introduced the research status and progress of Eurasian hydrology in this part, and the corresponding shortcomings were also pointed out.

[Figure]

—— Section "Materials and methods" I recommend to add a map to show the locations of the river basins, which could be a supplement to table 1, and I think it could effectively enhance the understanding of the study. In addition, more data descriptions should be added in the section, such as evaluation of the data quality.

—— Section "The results" The analysis in this part is credible, but the descriptions or presentation in some sentences should be revised according to the Figures. For example, there has some data lose in Figure 10, would it impact the trend conclusion of that part? The authors should detailed analyse and clear it in the text.

In section 3.4, the energy and water balance was considered to analyse the reason for river discharge variations. I suggest add trend lines in corresponding Figures or address some words in text to clear the trends of each energy budget. I think it would help to understand the discussion of this section.

——In addition, I've made some corrections of English writing, the major ones, but the manuscript still needs throughout edit on English.

Some edits:

Line 30: replace "nutrients" with "nutrient"

Line 49, 51 and throughout the text: replace "analyze" with "analyse", and replace "analyzed" with "analysed" to make the writing consistent in the whole text

Line 61: replace "kilometers" with "kilometres"

Line 64: replace "analyzing" with "analysing"

Line 93: should read SPSS (Statistical Product and Service Solutions) software

Line 105,108,111, 189, 195, and Table 1: replace "altitudes" with "latitudes"

Line 108: replace "R2" with "R2"

Line 150: should read "freeze-thaw"

Line 177: replace "Yeniesei" with "Yenisei"

Line 197: replace "Rs" with "Hs"

Line 206, 235, and 245: should read "significantly"

Line 209: should read "Rn, mainly"

Line 446: replace "that" with "which"
* * *

---

## Author Comment (AC1) · 8 May 2020

1. The English is poor, enough to reduce significantly the effectiveness of the discussion. Substantial editing would be needed – not just in terms of correcting individual words but also in terms of the phrasing of arguments. I had a very difficult time getting through some of the longer paragraphs in the results and discussion sections.

Reply: Thanks for your comments and suggestions. We agree with the opinion. We seriously revised the manuscript and have asked for an editor of a professional translation services company to check and revise the English of the manuscript. All the corresponding revisions will be marked in the manuscript.
2. Much of the data used here is not described well, and appropriate caveats or qual-ifications are missing. By not discussing the weaknesses of the datasets, the authors are essentially implying that they are all accurate enough for the analyses performed. This may or may not be true. a. What is the rain gauge density underlying the precipitation product? The gauge availability in high latitudes would probably be insufiňĄcient for particularly accurate data. Caveats are needed. b. The SWE data are said to come from the Terrestrial Water Budget Data Archive. Some description is necessary. Is this a model product? SWE is notoriously difiňĄcult to measure from space, and in situ measurements are presumably not comprehensive. Model products (including reanal-yses) are going to be error prone as well, so it's hard to believe that available data will be highly accurate. Again, qualification is needed. c. Data for surface energy budget terms and for soil moisture and temperature are taken from daily reanalysis. I see from the website quoted that these are Interim data rather than ERA5 data; this should be spelled out. Also, appropriate caveats regarding the accuracy of these data, particularly in areas that aren't well measured, are needed.

Reply: We fully agree with you and revised the questions one by one in the manuscript (in red). a. The rain gauges density in the study area and the accuracy of precipitation product has been added in the manuscript (Line 111-116). b. The SWE calculations and the data evaluation were added in the line 85-90. c. The description of ERA-Interim data and some caveats were added in the line 92-97.

3. The concept of "trend" is used too loosely in this paper. Typically a trend refers to somewhat consistent changes over multiple decades. Despite what's stated on lines 109-111, I don't see any obvious trend for MD and HD in high latitudes (certainly not a statistically significant trend), and in low latitudes, a trend is seen for MD only if the iňĄnal ten years are included – before that, there's no trend at all (i.e., for all we know, we could be looking at decadal-scale variability). (Also, it's very strange that the middle latitude MD and HD time series is described as a"slight decline"when this trend seems so much stronger than that for the high latitude MD and HD time series.) Line

123 says that the speed of ice-period shortening during 1996-2012 was intensified compared to that happening earlier, but from Figure 2, except for the Yana, it's not obvious that there was a statistically significant trend before 1996. By eye, it's not clear that there's any significant trend in Figure 7, despite the statement on line 144 – plenty of decadal variability, though. Again, talking in terms of trend rather than decadal variability is perhaps reading too much into the data. In any case, significance testing for trends is needed throughout. (I did see some p values listed here and there.)

Reply: Thanks for the comments and suggestions. We have revised the corresponding inappropriate sentences or words throughout the text. The significance testing also was added. The specific revisions were in red in manuscript (lines 131-134; lines 146-147; lines 167-173; lines 227-231)

4. The discussion on p. 7 (lines 190-218) was especially difficult to wade through, and I wasn't especially convinced by the arguments – for me, they came off as convoluted. I read through this text several times and am still not clear on the arguments. The discussion regarding permafrost, for example, comes off as unnecessary speculation. Perhaps breaking the paragraph down into more digestible bites would be useful. Here the concept of trend seems especially unclear, with discussion, for example, of an increasing LH trend up to 2000 and a decreasing trend thereafter (lines 216-217). It makes sense to talk about this in terms of decadal variability, not trends. (Also, in Figure 10, why are there breaks in the data in 1994 and 2009? Aren't the reanalysis data complete?)

Reply: we have rewritten this part, and split it into two paragraphs. The analysis was focus on decadal variability, and the unreasonable statements about "trend" were eliminated in the text. The data in 1994 and 2009 in Figure 10 was not complete and abnormal. We have added some explanations in the section "Materials and methods" (lines 95-97).

Minor points – A map is needed to show where the river basins are located and which

ones are included in the high/middle/low latitude categories. This will help the reader understand how representative these basins are for Eurasia in general.

Reply: Thanks for the suggestion. Table 1 listed the locations of the controlled hydrological stations of the eleven major river basins in Eurasia. They are categorized in different latitudinal zones, and the longitudes of the river basins (controlled hydrological stations) in different latitudinal zones were listed from west to east in Table 1. Considering Table 1 and the familiarity of these rivers, after seriously discussion, we decided not to add an additional location map in the manuscript.

– People will be confused by the term "Arctic-few discharge" in the abstract and on line 290.

Reply: Thanks. We agree with you and have revised it in the abstract and the conclusion.

– Line 73: Is the CERA-20C dataset really a climatology dataset, or do the values vary from year to year? I assume the latter. Are these reanalysis data?

Reply: CERA-20C is the ECMWF 10-member ensemble of coupled climate reanalyses of the 20th century. We have done the corresponding revisions in lines 75-77.

– Line 78: It probably should be stated explicitly here that sublimation impacts on the snow water balance are neglected. This could be a questionable assumption, as estimates in the literature seem to suggest that sublimation accounts for âĹij10-20% of the snow water balance.

Reply: thanks for the suggestion, and we have revised the text accordingly.

– Line 93: What is SPSS software?

Reply: the full name of SPSS is Statistical Product and Service Solutions, and we have done the revision in the text (line 107).

– Line 105 and throughout the text: "altitudes" to "latitudes"? Reply: they have been

revised in the text.

– I don't understand the slope ratio calculation on lines 165-166. Why not just look at RËȨ2 to determine how river ice to winter LD covary? This would make a lot more sense. The same comment applies to the calculation described on line 178.

Reply: Thanks for the suggestions. It is true that people always use the R2 to assess the impact of one factor to another, while in this paper, we believe that the contribution calculations according to slope ratio of river-ice and LD, as well as slope ratio of snowmelt water and HD, could more specifically describe the meaning.

– Line 197: Rs to Hs? Reply: it has been revised in the text.

– Lines 204-205: I don't understand this sentence; there must be a typo. Reply: have been revised the sentence (lines 225-226)

---

## Author Comment (AC2) · 8 May 2020

This manuscript analyzed the variations and the zonal differences of hydrological elements in Eurasia, and their causes during the recent few decades. I found the analysis results are interesting and believe that it would add useful information on water resource changes in middle and high latitudes of Northern Hemisphere. However, in my opinion, some revisions to the manuscript are needed, most of which are editorial, but also including the addition of some explanatory text.

– Section "Introduction" It well introduced the research status and progress of Eurasian hydrology in this part, and the corresponding shortcomings were also pointed out.

– Section "Materials and methods" I recommend to add a map to show the locations of the river basins, which could be a supplement to table 1, and I think it could effectively enhance the understanding of the study. In addition, more data descriptions should be added in the section, such as evaluation of the data quality.

Reply: Thanks for the suggestion. We agree and some data descriptions and evaluations were added in the section"Materials and methods" in lines 75-77, lines 82-90, lines 92-97, and lines 111-116. Table 1 listed the locations of the controlled hydrological stations of the eleven major river basins in Eurasia. They are categorized in different latitudinal zones, and the longitudes of the river basins (controlled hydrological stations) in different latitudinal zones were listed from west to east in Table 1. Considering Table 1 and the familiarity of these rivers, after seriously discussion, we decided not to add an additional location map in the manuscript.

– Section "The results" The analysis in this part is credible, but the descriptions or presentation in some sentences should be revised according to the Figures. For example, there has some data lose in Figure 10, would it impact the trend conclusion of that part? The authors should detailed analyse and clear it in the text.

Reply: Thanks for the suggestions. We have read through the manuscript and revised the sentences or words which were unclear and confused (Lines 131-134; Lines 216-218; Lines 221-222; Lines 227-232). The default data in 1994-1995 and 2009-1010 of Figure 10 were abnormal, and we have added some explanations in the section "Materials and methods" (Lines 95-97).

–In section 3.4, the energy and water balance was considered to analyse the reason for river discharge variations. I suggest add trend lines in corresponding Figures or address some words in text to clear the trends of each energy budget. I think it would help to understand the discussion of this section.

Reply: Thanks for the suggestions. We have rewritten this part, and split it into two paragraphs. The analysis was focus on decadal variability, and the unreasonable statements about "trend" were eliminated in the text. All the revisions were in red (Lines 216-240).

–In addition, I've made some corrections of English writing, the major ones, but the manuscript still needs throughout edit on English. Some edits: Line 30: replace "nutrients" with "nutrient" Line 49, 51 and throughout the text: replace "analyze" with "analyse", and replace "analyzed" with "analysed" to make the writing consistent in the whole text Line 61: replace "kilometers" with "kilometres" Line 64: replace "analyzing" with "analysing" Line 93: should read SPSS (Statistical Product and Service Solutions) software Line 105,108,111, 189, 195, and Table 1: replace "altitudes" with "latitudes" Line 108: replace "R2" with "R2" Line 150: should read "freeze-thaw" Line 177: replace "Yeniesei" with "Yenisei" Line 197: replace "Rs" with "Hs" Line 206, 235, and 245: should read "significantly" Line 209: should read "Rn, mainly" Line 446: replace "that" with "which"

Reply: we have accepted all the comments and revised them one by one in the manuscript. In addition, we have asked a professional English editor to check and revise English throughout the text. All revisions will be marked in the manuscript.

―――――――――――――――――――